# Pneumococcal Carriage in Infants Post-PCV10 Introduction in Pakistan: Results from Serial Cross-Sectional Surveys

**DOI:** 10.3390/vaccines10060971

**Published:** 2022-06-18

**Authors:** Shahira Shahid, Amala Khan, Muhammad Imran Nisar, Farah Khalid, Muhammad Farrukh Qazi, Sheraz Ahmed, Furqan Kabir, Aneeta Hotwani, Sahrish Muneer, Syed Asad Ali, Cynthia G. Whitney, Anita K. M. Zaidi, Fyezah Jehan

**Affiliations:** 1Department of Pediatric and Child Health, Aga Khan University, Karachi 74800, Pakistan; shahira.shahid.aku@gmail.com (S.S.); amala.khan@aku.edu (A.K.); farah.khalid@aku.edu (F.K.); muhammad.farrukh@aku.edu (M.F.Q.); sheraz.ahmed@aku.edu (S.A.); furqan.kabir@aku.edu (F.K.); aneeta.hotwani@aku.edu (A.H.); sehrish.munir@aku.edu (S.M.); asad.ali@aku.edu (S.A.A.); anita.zaidi@aku.edu (A.K.M.Z.); fyezah.jehan@aku.edu (F.J.); 2Emory University, Atlanta, GA 30322, USA; cynthiagwhitney@gmail.com; 3Bill & Melinda Gates Foundation, Seattle, WA 98102, USA

**Keywords:** streptococcus pneumoniae, 10-valent pneumococcal conjugate vaccine, Pakistan

## Abstract

The 10-valent pneumococcal vaccine was introduced in Pakistan’s Expanded Program on Immunization (EPI) in 2013 as a 3 + 0 schedule without catchup. We conducted three annual cross-sectional surveys from 2014–2016 to measure vaccine-type (VT) carriage in infants from a rural part of Pakistan. Nasopharyngeal specimens were collected by random sampling of infants from two union councils of Matiari. Samples were then transported to the Infectious Disease Research Laboratory (IDRL) at the Aga Khan University within 6–8 h of collection. Serotypes were established using sequential multiplex PCR. Of the 665 children enrolled across three surveys, 547 were culture-positive for pneumococcus. VT carriage decreased from 21.8% in 2014 to 12.7% in 2016 (*p*-value for trend <0.001). Those who were not vaccinated or partially vaccinated were found to be at higher risk of carrying a VT serotype ((aOR 2.53, 95% CI 1.39, 4.63 for non-vaccinated) and (aOR 3.35, 95% CI 1.82, 6.16 for partially vaccinated)). On the other hand, being enrolled in the most recent survey was negatively associated with VT carriage (aOR 0.51, 95% CI 0.28, 0.93). We found that PCV10 was effective in decreasing the carriage of vaccine-type serotypes in Pakistani infants.

## 1. Introduction

Pneumonia is a major cause of childhood morbidity and mortality worldwide. It has several etiologies of which *Streptococcus pneumoniae* (pneumococcus) is a leading one [1]. Pneumococci are Gram-positive bacteria, accountable for a wide spectrum of diseases both invasive (including pneumonia, bacteremia, and meningitis) and non-invasive [2].

Globally, the incidence of invasive pneumococcal disease in children under 5 years of age was about 14.5 million cases (uncertainty range 11.1–18.0 million) in 2000, resulting in almost 826,000 (582,000–926,000) deaths [3]. During the Millennium Development Goal (MDG) years of 2000–2015, the pneumococcal disease burden decreased globally, but its incidence increased in third world countries [4]. In 2015, Pakistan accounted for 7.1 million (4.2–12.0 million) pneumococcal pneumonia cases and around 63,960 pneumonia deaths in children less than 5 years of age [4].

There are more than 100 known serotypes of pneumococcus, which asymptomatically colonize the human respiratory tract. Out of these, only a few serotypes cause the majority of the disease [5,6]. Two Pneumococcal Conjugate Vaccine (PCV) formulations are currently WHO-prequalified for use in children: a 10-valent PCV (GSK, Synflorix^TM^) containing serotypes 1, 4, 5, 6B, 7F, 9V, 14, 18C, 19F, and 23F, and a 13-valent PCV (Pfizer, Prevenar 13^®^) containing three serotypes, 3, 6A, and 19A, in addition to PCV10 serotypes. Both PCV10 and PCV13 have proven to be the cornerstone of prevention against invasive pneumococcal disease worldwide [7,8,9,10]. Until 2020, PCVs had been introduced in 146 countries out of which in 60 countries it had been introduced with the help of Gavi, The Vaccine Alliance [11]. Another 10-valent PCV (PNEUMOSIL, Serum Institute of India, SIIPL-PCV) targeting serotypes 1, 5, 9V, 14, 19A, 19F, 23F, 7F, 6A, and 6B has recently been WHO pre-qualified [12,13]. Other higher valency PCV products (15-valent, 20-valent, and pan-serotype vaccines) are at various phases of development [14,15].

Pakistan was the first South Asian country to introduce PCV10 in its Expanded Program on Immunization (EPI) with the assistance of Gavi, The Vaccine Alliance in 2012 [16]. To match the existing immunization schedule, PCV10 was introduced as a 3 + 0 regimen, with doses at 6, 10, and 14 weeks of life without catch-up immunization [7,17]. The ten serotypes present in PCV10 are thus referred to as vaccine type (VT) serotypes.

As more data became available on prevailing serotypes in carriage and disease isolates in south Asia, PCV10 was replaced with PCV13 in the EPI of Pakistan in early 2021 [18]. This data-driven policy decision illustrates the importance of the continuous monitoring of prevailing serotypes after the introduction of vaccines. Since invasive disease has become rare in the post-vaccine scenario, nasopharyngeal carriage can provide important insights into the changing pneumococcal sero-epidemiology [19]. Thus, we carried out a series of cross-sectional surveys in two union councils of the rural Matiari district in the Sindh province, where we had nasopharyngeal carriage estimates available from the time before the introduction of the vaccine. The objectives were to measure the direct effect of conjugate vaccine on the prevalence of vaccine type serotypes in children who received the vaccine, as well as to measure the indirect effect in children who did not receive the vaccine. We also looked at factors associated with overall and VT carriage.

## 2. Materials and Methods

### 2.1. Study Design and Setting

This was a series of cross-sectional surveys conducted at yearly intervals for three years after the introduction of PCV10 in two union councils of rural Matiari (Khyber and Shah Alam Shah Jee Wasi), in the southern province of Sindh, Pakistan. A union council is the smallest administrative unit in the government structure. These sites were selected because the baseline study before vaccine introduction was done in the same population. Matiari is about 180 km away from Aga Khan University (AKU) in Karachi, where the research unit and the Infectious Disease Research Laboratory (IDRL) are located. Samples were collected from 1 January to 2 February in 2013 (phase 1, pre-PCV introduction), 14 February to 14 March 2014 (phase 2), 17 January to 3 March 2015 (phase 3) and 23 January to 14 March 2016 (phase 4). All the samples were collected approximately at the same time every year to account for seasonal variation in carriage rates. A random sample was drawn from a pre-existing line listing of children of selected union councils. All children aged 3 to 12 months residing in the study area were eligible to be enrolled.

Those excluded from the study were children with a history of severe acute respiratory illness in the last 2 weeks, the presence of one or more of the following symptoms: presence of chest wall indrawing, blue skin discoloration (cyanosis) and fast breathing, children with moderate-to-severe cerebral palsy, children with neurological disorders affecting swallowing, and those with nose and throat disorders. If a child selected from the random list was not found or not eligible, the next eligible child on the list was approached.

We updated the tool used in the baseline carriage survey to include information regarding pneumococcal vaccines and any other subsequent vaccines introduced in the Expanded Program on Immunization (EPI) since then. A two-day training session was held for staff involved in the study to standardize data and specimen collection methods.

### 2.2. Laboratory Methods

For the purpose of standardization, we used the same laboratory methods as in the base-line carriage survey [20]. Briefly, nasopharyngeal swabs (MW173P Transwab^®^ Pernasal Amies Plain, Wales, UK) were collected by trained staff following World Health Organization methods and immediately placed in skim milk tryptone glucose glycerol (STGG) media (Karachi, Pakistan) [17]. These were then transported at 2–8 °C within 6–8 h of collection to IDRL at AKU in Karachi, where the samples were frozen upright at −80 °C until further processing. Batches of 20–40 samples were thawed, cultured, and sub-cultured as mentioned in the CDC protocol [21].

Isolates were tested using conventional multiplex PCR to identify different pneumococcal serotypes [22]. DNA extraction was performed using the crude boiling method. It included boiling the bacterial culture for 10 min and centrifugation (Hitachi, Chiyoda, Tokyo, Japan) for another 10 min; the resulting supernatant was collected in a sterile microcentrifuge tube (Thermo Scientific, Waltham, MA, USA) and the remaining DNA extract was used for PCR techniques. The *cpsA* gene as an internal positive control was added in all multiplex reactions. An amount of 2 μL of DNA was added to the PCR master mix containing nuclease-free water, 2X Qiagen multiplex PCR buffer, Qiagen Q solution (Qiagen Multiplex Kit (Qiagen cat # 206145, Hilden, Germany), and 25 μM working stock of primers (Eurofins MWG operon, Luxembourg city, Luxembourg).

Amplification was carried out in an Eppendorf Master Cycler Gradient, Hamburg, Germany, with the specific temperature profile, and the amplified PCR products were stained and read under a BioRad Gel Doc imager (BioRad Universal Hood II, Hercules, CA, USA). Serotypes were detected through sequential multiplex conventional PCR, which were further confirmed by monoplex PCR, using the same PCR conditions, to avoid misidentification due to non-specific bands. Serogroup 6 was additionally differentiated into serotypes 6A, 6B, 6C, and 6D by same method used by Jin et al. with some modifications as mentioned earlier [23]. Pneumococcal serotype controls added in each reaction were obtained from the CDC streptococcal lab. The non-typeable products were confirmed by pneumococci using *lytA* real time PCR per Carvalho et al. [24].

For quality control in the Optochin susceptibility and bile solubility reactions, *Streptococcus pneumoniae* ATCC 49619 and *Enterococcus faecalis* ATCC 29212 strains of American Type Culture Collection (ATCC) were used.

### 2.3. Sample Size

The sample size was calculated based on a predicted decline in nasopharyngeal carriage in upcoming years after the introduction of the vaccine. There were no recent data regarding the prevalence of pneumococcal carriage in Pakistan, thus the sample size was calculated on the basis of previously done surveys. To obtain the maximum sample size, a baseline carriage rate of 50% was assumed. The sample size was calculated to be 220 for each round of annual surveys in order to show a decline in prevalence of 0·2 (from 0.5 to 0.3) after the introduction of the ten-valent vaccine with a 5% level of significance and 80% power.

### 2.4. Data Management and Quality Assurance

The questionnaire consisted of demographics, socio-economic indicators, and general health status and was collected on paper forms by trained staff at the appointed sites. All children who participated were given a unique identifier which was linked to an identification number in the demographic surveillance system. All completed forms were reviewed by the study supervisor at the end of the day for completeness and consistency.

PCV10 carriage was defined as isolation of any of the 10 serotypes included in PCV10 (serotypes 1, 4, 5, 6B, 7F, 9V, 14, 18C, 19F, 23F). Non-vaccine type (NVT) carriage was defined as presence of all other pneumococci including the non-typeable. PCV13 serotypes were defined as any of the PCV13 specific serotypes 3, 6A, and 19A. Infants who received all three PCV10 doses were considered as fully vaccinated, those who received 1 or 2 doses were considered partially vaccinated, and those with no doses were considered unvaccinated. Data entry was done using RedCap software, on which validation and consistency checks were run. Analysis was done using STATA version 16.0. The general characteristics of study participants are presented as mean (SD) or median (IQR) based on normality for quantitative variables and numbers with percentages for categorical variables. For each survey, we estimated the overall prevalence of pneumococcal carriage, PCV10 carriage, PCV13 carriage, and non-vaccine serotypes with 95% confidence intervals (CIs). Logistic regression was used to identify predictors associated with carriage. All variables found to be significant at a *p*-value of ≤0.10 at the univariate level were added to the multivariate model· A stepwise backward elimination approach was used to derive a parsimonious model, retaining only variables with a *p*-value of <0.05. Odds ratios with 95% confidence intervals are presented.

## 3. Results

We conducted a series of cross-sectional surveys for three consecutive years (2014–2016) after PCV10 introduction. An earlier published pre-PCV10 introduction survey conducted in 2013 in the same population and using the same methodology served as the baseline for comparison [20]. In total 224, 221, and 220 children participated in the 2014, 2015, and 2016 surveys respectively. The baseline demographics and clinical characteristics of enrolled children are presented in Table 1. The overall mean age of the participants was 7.24 ± 2.75 months, the majority of the children were male, and the majority of the primary caretakers 644 (84%) had no education at all; the median crowding index was 5.5 (4–7.5); around half of children were exposed to environmental tobacco smoke (ETS).

All samples collected were of sufficient quality for analysis. Distributions of the most prevalent VT and NVT serotypes for each year are presented in Figure 1 and Figure 2. Total pneumococcal carriage remained relatively stable over time (range 79.5 to 81.8%), with no significant reduction noticed (*p*-value for trend 0.340 Table 2).

However, VT carriage decreased from 21.8% in 2014 to 12.7% in 2016 (*p*-value for trend <0.001). A gradual decrease in prevalence of the three PCV13 specific serotypes 3, 6A, and 19A was also noticed as a group, but this was not statistically significant (*p*-value for trend 0.141). However, the prevalence of NVT carriage increased significantly from 37.3% in 2014 to 57.2% in 2016 (*p*-value for trend <0.001). At the same time, the proportion of fully vaccinated children (per verbal report or card verification) increased from 14.2 to 48.1% (*p*-value < 0.001). VT carriage in fully vaccinated children decreased from 9.3 to 7.5% (*p*-value < 0.001, as depicted in Table 3 and Table 4), whereas for unvaccinated children, VT carriage decreased from 25.1 to 15.6% (*p*-value for trend <0.001).

Table 5 describes the association of different characteristics with a positive nasopharyngeal culture. In the final adjusted multivariate model, increasing age in months was found to be significantly associated with culture positivity (aOR; 1.08, 95% CI 1.01, 1.16), whereas children with two or more outpatient visits were less likely to have a positive culture (aOR; 0.54, 95% CI 0.35, 0.89).

Table 6 shows the association of these characteristics with VT carriage. Those who were not vaccinated and partially vaccinated were found to be at higher risk of carrying a VT serotype when compared with fully vaccinated children ((aOR; 2.53, 95% CI 1.39, 4.63) and (aOR; 3.35, 95% CI 1.82, 6.16), respectively). On the other hand, being enrolled in the last survey was negatively associated with VT carriage (aOR; 0.51, 95% CI 0.28, 0.93) (Table 6).

## 4. Discussion

In our study we saw a significant decline in the point prevalence of pneumococcal Vaccine Type (VT) carriage after three years of PCV10 introduction in a previously vaccine-naïve rural population. When compared to the pre-introduction VT carriage rate of 26.6% in the same population, a greater than 50% decline was observed. This decline was more pronounced in the vaccinated group when compared to the partially/non-vaccinated group. However, a significant decline in the non-vaccinated group after around half of the children in the community were vaccinated hints towards an indirect effect of the vaccine.

Similar effects have been seen in populations across the world. Another study done over four years (2014–2018) in the same rural population of Matiari, Pakistan, which enrolled 3140 children younger than 2 years and followed a slightly different study design showed a decline of greater than 50% in VT carriage, from 16.1% in 2014/15 to 9.6% in 2017/18 [25]. A four-year survey done in Fiji, from 2012 (pre-vaccination introduction) to 2015 (post vaccine introduction), showed a 63% reduction in PCV10 serotype carriage in children aged 12–23 months. The percentage of participants who were PCV10-vaccinated increased from zero in 2012 to 2% in 2013 to 95% in 2014 to 100% in 2015 in this study. The researchers also observed a decrease in VT carriage in older unvaccinated children and infants too young to be vaccinated [26]. A series of four annual cross-sectional study done in Kilifi, Kenya, two were done before the introduction of PCV10, and two were done afterwards, which showed that the carriage of VT serotypes dropped two-thirds in both children and older individuals after vaccine coverage reached 79%. This study was a milestone, as these findings suggested that PCV10 introduction can have a substantial indirect effect on the unvaccinated population [27]. PCV10 was introduced in 2010 in Brazil, and cross-sectional surveys done in 2010 and 2013 showed decreases in vaccine type pneumococcal carriage from 19.8 to 1.8% at a vaccine coverage of 95% [28]. All these studies, done in different parts of the world, in different ethnic populations and distinct environments, have shown large decreases in vaccine-type pneumococcal carriage in both vaccinated and un-vaccinated population· Because carriage is a precursor of disease, we can reasonably infer that these decline in carriage rates would translate to a decline in disease.

Another important finding from our study is that the overall pneumococcal carriage remained similar throughout. It was the composition of the serotypes that changed from a predominantly vaccine type to a predominantly non-vaccine type. In baseline samples, we see a higher prevalence of vaccine-type serotypes, but after vaccine introduction these serotypes have been replaced by non-vaccine type (NVT) serotypes, keeping overall carriage rate the same. This phenomenon, known as serotype replacement, has also been observed in other parts of the world. A study done in the Netherlands showed a large reduction in vaccine serotypes after pneumococcal vaccination but also immediate and complete replacement by non-vaccine pneumococcal (NVT) serotypes [29]. A study done in Kenya five years after introduction of a vaccine showed a 76% prevalence of pneumococcal carriage in children under 5 years [27] and another study in Mozambique two years after the introduction of PCV10 reported an overall carriage rate of 84.8% in children under 5 years of age [27,30].

In our study we assessed predictors that are believed to influence pneumococcal carriage, such as age, gender, previous hospital admissions, outpatient visits, and a patient’s exposure to passive smoking. In the regression modeling, increasing age and number of outpatient visits in the last year were found to be associated with overall carriage.

Existing scientific literature shows varied results regarding factors associated with carriage; some studies agree with our findings while others showed no correlation with carriage and these predictors. A study conducted in the Netherlands on the predictors of pneumococcal colonization also found a decline in colonization with increasing age, which may be related to maturation of immune system with age [31]. Ghaffar et al. showed that passive smoking at home, large households, and crowding among teenagers were correlated with higher pneumococcal carriage rates [32]. Another study done in the Netherlands also examined predictors of pneumococcal carriage, but failed to show any correlation with specific factors [33]. Similar findings were reported from a study done in Cyprus, which showed no association with previous antimicrobial use or history of upper respiratory infection [34]. On the other hand, some studies have found a negative effect of previously treated respiratory tract infection/antimicrobial use on carriage rate [35,36].

The introduction of vaccine directly benefited those who got the vaccine and indirectly benefited those who didn’t get the vaccine. This is consistent with the herd effect of immunization as exemplified by the Halloran model [37]. Vaccination status was found to be negatively associated with VT carriage. The progressing years of the study were also found to be negatively associated with VT carriage rate, which is again consistent with the spilling over of beneficial effects of vaccine from the vaccinated to unvaccinated group. A four-year cross-sectional survey done in Fiji found PCV-10 carriage to be negatively associated with vaccination status and the year of enrollment in the study. In the same study, predictors such as young age, urban residence, living with two or more children under five years, low family income, symptoms of URTI, and exposure to household cigarette smoke were found to be positively associated with VT carriage [38]. A study done in Brazil, 3 years after the introduction of PCV10, showed that VT carriage declined by >90%; with a reduction of 87.8% among partially vaccinated patients and 97% in the fully vaccinated ones [28]. Thus, in light of the above findings and consistent literature, we can infer that both vaccine coverage in the community and the number of years since the introduction of the vaccine have a profound effect on VT carriage.

One of the strengths of our study was the availability of baseline carriage data as a comparison group. We performed all surveys during the same months every year to account for seasonal variations in carriage. Our study had several limitations. Vaccination cards were not available for around 40% of the vaccinated children, thus we relied on parental recall for vaccination status. Since our study was performed in one rural district of Pakistan, the carriage rates may not be fully representative of other peri-urban and urban settings. Nevertheless, these results provide evidence of the effect of PCV10 on carriage in the same population over time which we believe would be replicable in other parts of Pakistan. Moreover, in the context of PCV13 introduction in Pakistan’s EPI in early 2021, our study provides invaluable baseline data for comparison with future surveys.

## Figures and Tables

**Figure 1 vaccines-10-00971-f001:**
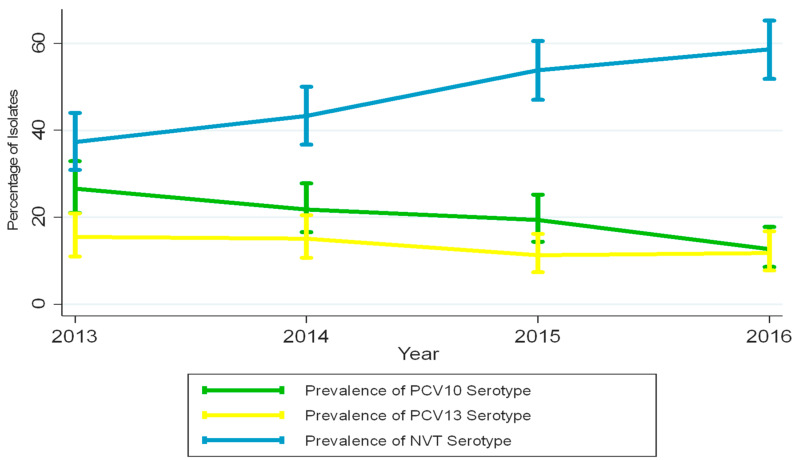
Proportion of children carrying PCV10, PCV13, and NVT serotypes over time. Rates for 2013 are from a previously published survey in the same community by the same group of authors.

**Figure 2 vaccines-10-00971-f002:**
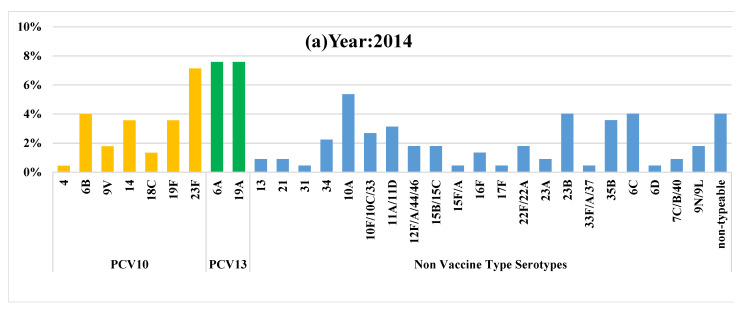
Prevalence of pneumococcal serotypes across three surveys (**a**) 2014, (**b**) 2015. and (**c**) 2016 in children 3–12 months old in Matiari, Pakistan.

**Table 1 vaccines-10-00971-t001:** Sociodemographic characteristics of enrolled children.

Characteristics	2014	2015	2016
	*n* = 224	*n* = 221	*n* = 220
Age, months (Mean ± SD)	7.1 ± 2.60	7.5 ± 2.7	7.8 ± 2.94
Gender			
Male	108 (48.2%)	113 (51.1%)	113 (51.6%)
Female	116 (51.8%)	108 (48.9%)	106 (48.4%)
Education of Caretaker			
Illiterate	150 (81.1%)	119 (84.4%)	184 (84.8%)
1–5 Years	25 (13.5%)	17 (12.1%)	25 (11.5%)
6–16 Years	10 (5.4%)	5 (3.5%)	8 (3.7%)
Education of Primary wage earner			
No Education	122 (54.5%)	76 (42.0%)	128 (59.0%)
1–5 years	34 (15.2%)	61 (33.7%)	46 (21.2%)
6–16 years	68 (30.4%)	44 (24.3%)	43 (19.8%)
People in the household, Median (IQR)	8 (6–11.5)	8 (6–11)	8 (6–11)
No. of rooms in house, Median (IQR)	1 (1–2)	1 (1–2)	1 (1–2)
Crowding index, * Median (IQR)	6 (4–8)	5 (4–7)	5.5 (4–8)
Hospital Admissions in the last year			
None	206 (92.0%)	207 (93.7%)	217 (99.1%)
One	16 (7.1%)	12 (5.4%)	1 (0.5%)
Two or more	2 (0.9%)	2 (0.9%)	1 (0.5%)
Outpatient visits in the last month			
None	118 (52.7%)	109 (49.5%)	166 (75.8%)
One	66 (29.5%)	58 (26.4%)	18 (8.2%)
Two or more	40 (17.9%)	53 (24.1%)	35 (16.0%)
Smoker in household			
Yes	136 (60.7%)	101 (45.7%)	101 (46.1%)
No	88 (39.3%)	120 (54.3%)	118 (53.9%)
Exposed to smoke during cooking			
Yes	179 (79.9%)	47 (21.3%)	66 (30.1%)
No	44 (19.6%)	174 (78.7%)	153 (69.9%)
Don′t Know	1 (0.4%)	0 (0.0%)	0 (0.0%)
Ever Vaccinated			
Yes	184 (82.1%)	165 (74.7%)	184 (83.6%)
No	40 (17.9%)	56 (25.3%)	36 (16.4%)

IQR = Interquartile Range, * Defined as number of persons/number of rooms.

**Table 2 vaccines-10-00971-t002:** Vaccine type and non-vaccine type (NVT) carriage across three surveys in children 3–12 months of age in Matiari, Pakistan.

Year	2014	2015	2016
	(*n*= 224)	(*n* = 221)	(*n* = 220)
Positive for pneumococcus (*n*)	180	187	180
Prevalence of pneumococcus % (95% CI) *	80.3 (74.5–85.3)	84.6 (79.1–89.1)	81.8 (76.0–86.6)
Vaccine type serotypes (*n*)	49	43	28
Prevalence of vaccine type serotypes % (95% CI) ᵝ	21.8 (16.6–27.8)	19.4 (14.4–25.2)	12.7 (8.6–17.8)
NVT serotypes (*n*)	131	144	152
Prevalence of NVT serotypes % (95% CI) ᵞ	58.4 (51.7–65.0)	65.1 (58.4–71.4)	69.0 (62.5–75.1)

** p*-value for trend = 0.340, ᵝ *p*-value for trend < 0.001, ᵞ *p*-value for trend 0.020.

**Table 3 vaccines-10-00971-t003:** Vaccination status across the three surveys in children 3–12 months of age in Matiari.

PCV10 Vaccination Status	2014	2015	2016
Unvaccinated, % (95% CI) *	67.4 (60.8–73.5)	37.5 (31.1–44.3)	23.1 (17.7–29.3)
Partially vaccinated, % (95% CI) *	18.3 (13.4–24.0)	32.5 (26.4–39.1)	28.6 (22.7–35.0)
Fully vaccinated, % (95% CI) *	14.2 (9.9–19.5)	29.8 (23.9–36.3)	48.1 (41.4–54.9)

* *p*-value for trend <0.001 for all categories.

**Table 4 vaccines-10-00971-t004:** VT Carriage rates by vaccination status across the three surveys.

PCV10 Vaccination Status	2014	2015	2016
Unvaccinated, % (95% CI) *	25.1 (18.4–32.8)	16.8 (9.5–26.6)	15.6 (7.0–28.5)
Partially vaccinated, % (95% CI) *	19.5 (8.8–34.8)	31.9 (21.4–43.9)	19.0 (10.2–30.9)
Fully vaccinated, % (95% CI) *	9.3 (1.9–25.0)	9.0 (3.4–18.7)	7.5 (3.3–14.3)

* *p*-value for trend <0.001.

**Table 5 vaccines-10-00971-t005:** Factors associated with a positive nasopharyngeal culture among study participants.

Predictors	Culture Negative	Culture Positive	Unadjusted OR(CI)	Adjusted OR(CI)
	*n* = 118	*n* = 547		
Age, months (Mean ± SD)	6.9 ± 2.7	7.6 ± 2.7	1.1 (1.0–1.2)	1.1 (1.0–1.2)
Gender				
Male	66 (56.4%)	268 (49.0%)	Reference	
Female	51 (43.6%)	279 (51.0%)	1.3 (0.9–2.0)	
Education of Caretaker				
Illiterate	76 (77.6%)	377 (84.7%)	Reference	
1–5 Years	16 (16.3%)	51 (11.5%)	0.6 (0.3–1.1)	
6–16 Years	6 (6.1%)	17 (3.8%)	0.6 (0.2–1.5)	
Education of Primary wage earner				
No Education	59 (53.6%)	267 (52.1%)	Reference	
1–5 years	20 (18.2%)	121 (23.6%)	1.3 (0.7–2.3)	
6–16 years	31 (28.2%)	124 (24.2%)	0.8 (0.5–1.4)	
Total No. people in child′s house, Median (IQR)	8 (6–11)	8 (6–11)	1.0 (0.9–1.0)	
No. of rooms in child′s house, Median (IQR)	1 (1–2)	1 (1–2)	1.0 (0.9–1.0)	
Crowding index, Median (IQR)	5 (4–8)	5.5 (4–7.6)	1.0 (0.9–1.1)	
Hospital Admissions in the last year				
None	107 (91.5%)	523 (95.6%)	Reference	
One	8 (6.8%)	21 (3.8%)	0.5 (0.2–1.2)	
Two or more	2 (1.7%)	3 (0.5%)	0.3 (0.05–1.9)	
Outpatient visits in the last month				
None	56 (47.9%)	337 (61.7%)	Reference	
One	34 (29.1%)	108 (19.8%)	0.5 (0.3–0.9)	
Two or more	27 (23.1%)	101 (18.5%)	0.6 (0.4–1.0)	0.5 (0.4–0.9)
Smoker in household				
Yes	59 (50.4%)	267 (48.8%)	1.0 (0.7–1.6)	
No	58 (49.6%)	280 (51.2%)	Reference	
Exposed to smoke during cooking				
Yes	46 (39.3%)	246 (45.0%)	1.3 (0.8–1.9)	
No	71 (60.7%)	300 (54.8%)	Reference	
Don′t Know	0 (0.0%)	1 (0.2%)	-	
PCV10 Vaccination Status				
Non-Vaccinated	59 (50.0%)	226 (41.3%)	0.6 (0.4–1.0)	
Partially Vaccinated	30 (25.4%)	146 (26.7%)	0.8 (0.5–1.4)	
Fully Vaccinated	29 (24.6%)	175 (32.0%)	Reference	
Year of Enrollment				
2014	44 (37.3%)	180 (32.9%)	Reference	
2015	34 (28.8%)	187 (34.2%)	1.3 (0.8–2.2)	
2016	40 (33.9%)	180 (32.9%)	1.1 (0.7–1.8)	

**Table 6 vaccines-10-00971-t006:** Factors associated with the carriage of PCV10-serotype pneumococci among study participants.

Predictors	PCV10 Negative	PCV10 Positive	Unadjusted OR(CI)	Adjusted OR(CI)
	*n* = 710	*n* = 180		
Age, months (Mean ± SD)	7.6 (2.8)	7.0 (2.6)	0.9 (0.8–1.0)	
Gender				
Male	271 (49.8%)	63 (52.5%)	Reference	
Female	273 (50.2%)	57 (47.5%)	0.9 (0.6–1.3)	
Education of Caretaker				
Illiterate	365 (81.7%)	88 (91.7%)	Reference	
1–5 Years	60 (13.4%)	7 (7.3%)	0.5 (0.2–1.1)	
6–16 Years	22 (4.9%)	1 (1.0%)	0.2 (0.2–1.4)	
Education of Primary wage earner				
No Education	266 (52.3%)	60 (53.1%)	Reference	
1–5 years	113 (22.2%)	28 (24.8%)	1.1 (0.6–1.8)	
6–16 years	130 (25.5%)	25 (22.1%)	0.9 (0.5–1.4)	
Total No. people in child′s house, Median (IQR)	8 (6–11)	8 (6–11)	1.0 (0.9–1.0)	
No. of rooms in child′s house, Median (IQR)	1 (1–2)	1 (1–2)	1.0 (0.9–1.0)	
Crowding index Median (IQR)	5.3 (4–7.6)	6 (4.1–8)	1.0 (0.9–1.1)	
Hospital Admissions in the last year				
None	518 (95.2%)	112 (93.3%)	Reference	
One	22 (4.0%)	7 (5.8%)	1.5 (0.6–3.5)	
Two or more	4 (0.7%)	1 (0.8%)	1.2 (0.1–10.4)	
Outpatient visits in the last month				
None	323 (59.5%)	70 (58.3%)	Reference	
One	121 (22.3%)	21 (17.5%)	0.8 (0.4–1.3)	
Two or more	99 (18.2%)	29 (24.2%)	1.3 (0.8–2.2)	
Smoker in household				
No	265 (48.7%)	61 (50.8%)	Reference	
Yes	279 (51.3%)	59 (49.2%)	0.9 (0.6–1.3)	
Exposed to smoke during cooking				
No	308 (56.6%)	63 (52.5%)	Reference	
Yes	235 (43.2%)	57 (47.5%)	1.2 (0.7–1.7)	
PCV10 Vaccination Status				
Non-Vaccinated	225 (41.3%)	60 (50.0%)	2.9 (1.6, 5.1)	2.4 (1.2–4.8)
Partially Vaccinated	133 (24.4%)	43 (35.8%)	3.5 (1.9, 6.5)	2.8 (1.3–5.9)
Fully Vaccinated	187 (34.3%)	17 (14.2%)	Reference	Ref
Year of Enrollment				
2014	175 (32.1%)	49 (40.8%)	Reference	
2015	178 (32.7%)	43 (35.8%)	0.8 (0.5–1.3)	
2016	192 (35.2%)	28 (23.3%)	0.5 (0.3–0.8)	0.5 (0.3–1.0)

## Data Availability

The data presented in this study are available on request from the corresponding author.

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
