# Peer review of "Pneumococcal Carriage in Infants Post-PCV10 Introduction in Pakistan: Results from Serial Cross-Sectional Surveys"

_vaccines, 2022, doi:10.3390/vaccines10060971_

Round 1

Reviewer 1 Report

This work shows the results of cross- sectional surveys on pneumococcal carriage in Pakistani infants after the introduction of PCV10. Nasopharyngeal specimens were collected and treated in Aga Khan University (sequential multiplex PCR). After checking the results, the work shows that VT carriage decreased from 21.8% in 2014 to 12.7% in 2016. Unvaccinated/partially vaccinated children were at higher risk of carrying a VT serotype, which also demonstrated that the vaccination has a positive effect on decreasing the carriage of vaccine-type serotypes.

The work is not novel, since there are thousands of similar works being published in the last 10 years, but it important and interesting, particularly because it is related to a country that has recently introduced this vaccine.

There are, however, several details not well explained and need to be corrected/adjusted.

  • Replace dots with commas on the displayed numbers (percentages, total numbers...);
  • Kit for DNA extraction? (brand and country)
  • All materials and reagents need to have brand and country information. Several do not have this info. Please add;
  • Line 116: “...real time PCR per Carvalho at al” – et al. Please correct;
  • Table 1 has bold in all results. It makes it difficult to read. Please adjust;
  • Figure 1 (graph) is distorted. Please adjust;
  • Table 6: define “Ref”;
  • The last paragraph of the “Discussion” section could be re-written, as it is not very readable, but it is only an opinion.

Author Response

This work shows the results of cross- sectional surveys on pneumococcal carriage in Pakistani infants after the introduction of PCV10. Nasopharyngeal specimens were collected and treated in Aga Khan University (sequential multiplex PCR). After checking the results, the work shows that VT carriage decreased from 21.8% in 2014 to 12.7% in 2016. Unvaccinated/partially vaccinated children were at higher risk of carrying a VT serotype, which also demonstrated that the vaccination has a positive effect on decreasing the carriage of vaccine-type serotypes. The work is not novel, since there are thousands of similar works being published in the last 10 years, but it important and interesting, particularly because it is related to a country that has recently introduced this vaccine.

Thank you for the positive feedback.

There are, however, several details not well explained and need to be corrected/adjusted.

  • Replace dots with commas on the displayed numbers (percentages, total numbers...);

Thank you for the comment, we have revised the document accordingly.

  • Kit for DNA extraction? (Brand and country)

Thank you for the comment. We used crude boiling method for extracting DNA from pneumococcal isolates, no kit was used.

  • All materials and reagents need to have brand and country information. Several do not have this info. Please add.

Thank you for the comment. We have now added the brand and country information in the methods section. (Lines 95 to 118 of the revised manuscript)

  • Line 116: “...real time PCR per Carvalho at al” – et al. Please correct;

Thank you. We have corrected the typo in line 123 of the revised manuscript..

  • Table 1 has bold in all results. It makes it difficult to read. Please adjust;

Thank you. We have revised table 1 to improve the readability.

  • Figure 1 (graph) is distorted. Please adjust.

Thank you, we have revised figure 1.

  • Table 6: define “Ref”;

Thank you for the comment. We have now defined ‘ref’ wherever necessary.

  • The last paragraph of the “Discussion” section could be re-written, as it is not very readable, but it is only an opinion.

We agree with the reviewer and have revised the text for better clarity. (Lines 285-290 of the revised manuscript)

Reviewer 2 Report

Dear Editor and Authors,

It was my pleasure to review this study in which the authors have demonstrated the significant effect (more than 50% decline) that the introduction of the PCV10 vaccine for S. pneumoniae had over a three year period in the point prevalence of pneumococcal Vaccine Type (VT) carriage in a previously vaccine naïve rural population in two small administrative regions in Pakistan.

I must say this is a very well conducted study with a well thought out methodology and the authors have followed the principles of sound scientific research. They have clear inclusion and exclusion criteria, they have performed a sample size calculation and they have done an good analysis. Their findings are quite interesting, if maybe not so unexpected and are well presented and documented both with clear tables and graphs in the manuscript. The only limitation is the fact that they were limited as to the regions they could perform their surveys due to the unavailability of pre-vaccination data!

Overall I am quite satisfied by this work and would recommend it for publication. I congratulate the authors for their perseverance.

My kindest regards. 

Author Response

Dear Editor and Authors,

It was my pleasure to review this study in which the authors have demonstrated the significant effect (more than 50% decline) that the introduction of the PCV10 vaccine for S. pneumoniae had over a three-year period in the point prevalence of pneumococcal Vaccine Type (VT) carriage in a previously vaccine naïve rural population in two small administrative regions in Pakistan.

I must say this is a very well conducted study with a well thought out methodology and the authors have followed the principles of sound scientific research. They have clear inclusion and exclusion criteria; they have performed a sample size calculation and they have done a good analysis. Their findings are quite interesting, if maybe not so unexpected and are well presented and documented both with clear tables and graphs in the manuscript. The only limitation is the fact that they were limited as to the regions they could perform their surveys due to the unavailability of pre-vaccination data!

Thank you for the comments. We have included the generalizability of the results as a limitation (Lines 289-290 of the revised manuscript).

Overall, I am quite satisfied by this work and would recommend it for publication. I congratulate the authors for their perseverance.

My kindest regards. 

We thank the reviewer for the positive comments.

Reviewer 3 Report

Very well designed research, and the report well organized and written, giving a valuable contribution to our knowledge. There are just minor corrections to be done.

1. When, in the manuscript, bacteria Streptococcus pneumoniae was mentioned the first time, pneumococcus was put in the brackets meaning that the authors announce that this will be used later in the text when talking about Streptococcus pneumoniae, nevertheless the authors used the not-announced abbreviation S. pneumoniae. This has to be corrected.

2. The abbreviation “VT” in the Introduction chapter needs an explanation.

Author Response

Very well-designed research, and the report well organized and written, giving a valuable contribution to our knowledge.

Thank you for the constructive feedback.

There are just minor corrections to be done.

  1. When, in the manuscript, bacteria Streptococcus pneumoniae was mentioned the first time, pneumococcus was put in the brackets meaning that the authors announce that this will be used later in the text when talking about Streptococcus pneumoniae, nevertheless the authors used the not-announced abbreviation  pneumoniae. This has to be corrected.

Thank you for the comment. We have revised the text to include the correct abbreviation (Lines 39 and 260 of the revised manuscript).

  1. The abbreviation “VT” in the Introduction chapter needs an explanation.

Thank you for the comment. We have now included an explanation for ‘VT’ carriage in lines 55-56 of the introduction section in the revised manuscript.